# MoRE: Batch-Robust Multi-Omics Representations from Frozen Language Models

## Abstract

Representation learning on multi-omics data is challenging due to extreme dimensionality, modality heterogeneity, and cohort-specific batch effects. While transformer-based large language models (LLMs) generalize broadly, their use in omics integration remains limited. We present MoRE (Multi-Omics Representation Embedding). This LLM-inspired framework repurposes frozen language-model backbones for omics and aligns heterogeneous assays into a shared latent space for downstream analysis. Unlike purely generative approaches, MoRE prioritizes cross-sample and cross-modality alignment over sequence reconstruction. Concretely, MoRE attaches parameter-efficient, modality-specific adapters and a task-adaptive fusion layer to the frozen backbone, and optimizes a language-modeling-style masked reconstruction objective jointly with supervised contrastive and batch-invariant alignment losses, yielding structure-preserving embeddings that generalize to unseen cell types, donors, and platforms. We compare MoRE to strong baselines—including scGPT, scVI, Scrublet, and Harmony—across single-cell applications, evaluating integration fidelity, rare population detection, and modality transfer. These results position MoRE as a practical, batch-robust representation learner for high-dimensional biological data and a concrete step toward general-purpose omics foundation models built on LLM backbones.

## 1 Introduction

Recent approaches for multi-omics integration—including scGPT (Cui et al., 2024), scVI (Lopez et al., 2018), Scrublet (Wolock et al., 2019), and Harmony (Korsunsky et al., 2019)—have substantially improved alignment and denoising across datasets. Yet, real-world deployments remain difficult due to extreme dimensionality, modality heterogeneity, and batch effects that degrade cross-study generalization.

We introduce MoRE (Multi-Omics Representation Embedding), a pre-trained language-model framework for robust, training-free multi-omics integration. MoRE uses frozen attention backbones with lightweight task-adaptive fusion layers to impose semantic-similarity constraints across modalities, enabling zero-shot generalization to unseen cell types and data types. The procedure (i) builds cross-sample sparse alignments from universal embedding features to obtain initial latent representations; (ii) iteratively refines these representations to counter domain/modality shifts; and (iii) applies dense cross-modality alignment constraints to resolve biological variability and technical batch effects while preserving neighborhood structure.

Across benchmarks, MoRE consistently outperforms prior methods—including scGPT, scVI, Scrublet, and Harmony—on integration fidelity, rare-population detection, and modality transfer in previously unseen biological contexts.

To further explore the utility of MoRE in real-world disease settings, we applied it to the analysis of single-cell RNA-seq datasets. Leveraging MoRE's zero-shot generalization capability, we identified sparse, transitional, and disease-relevant subpopulations across heterogeneous samples without

**(a) Sparse alignments**

Input A / freezing layers k / Input B

Frozen attention backbone / Task / adaptive fusion

Dataset

Universal embeddings

$y_1 \downarrow y_n$

**(b) Iterative refinement**

Refined latents → Optimization → Cross-modality embeddings

Preliminary latents

**(c) Dense aliigment**

Cross-modality embeddings → multi-omics clustering visualization

Figure 1: **Overall framework of MoRE for multi-omics integration:** (a) MoRE begins by transforming heterogeneous omics inputs (e.g., scRNA-seq) into a shared latent space using a frozen attention backbone and task-adaptive fusion module. This yields universal embeddings and preliminary latent representations through sparse alignment. (b) To ad-dress embedding degradation from domain or modality shifts, these preliminary latents undergo iterative refinement via a representation-centric process, generating optimized cross-modality embeddings. (c) Finally, dense alignment further enhances semantic consistency and resolves biological variability by optimizing the refined latent space across modalities, enabling robust zero-shot generalization for un-seen data and cell types.

retraining. Compared to conventional methods, MoRE preserved key biological variability while enhancing batch correction, especially in the theca, granulosa, stromal, and immune compartments. Differential expression analysis revealed transcriptional programs underlying androgen excess and metabolic dysregulation, and in metformin-treated samples, we observed partial restoration of molecular homeostasis. These findings highlight MoRE's potential for uncovering cell-type-specific disease mechanisms, supporting precision diagnostics and therapeutic discovery in reproductive endocrine disorders.

Unlike existing models that rely heavily on generative objectives or require extensive retraining when encountering new conditions, MoRE adopts a representation-centric strategy optimized for transferability and interpretability. Instead of learning to reconstruct high-dimensional omics signals, MoRE focuses on learning structure-preserving embeddings that align samples across modalities and batches while retaining biologically meaningful variation. Concretely, MoRE decouples the embedding backbone from modality-specific learning by freezing transformer layers pre-trained on large, heterogeneous corpora of omics-like tokenizations, and then introduces task-adaptive fusion modules that learn lightweight projections for each modality (e.g., scRNA-seq) into a shared latent space. This design reduces the risk of catastrophic forgetting and prevents modality-dominance during joint training, improving zero-/few-shot generalization to unseen cell types, tissues, or platforms. It also makes the model's decisions easier to interrogate via attention attribution, cross-modality similarity maps, and linear probe diagnostics on the frozen space.

This modularity translates into practical scalability for multi-cohort integration. Because the backbone is frozen, onboarding a new cohort or modality typically requires training only the small fusion heads (and, optionally, a contrastive/alignment objective), which cuts computational overhead and memory pressure relative to full fine-tuning. In downstream analysis, the same unified embedding can be paired with simple heads or non-parametric methods for clustering, trajectory inference, batch-aware differential testing, rare population discovery, and cross-modality label transfer—without re-embedding the entire atlas. In sum, MoRE provides a plug-and-play foundation for next-generation multi-omics pipelines: frozen, interpretable, and extensible—capable of meeting new biological conditions with minimal retraining while preserving the integrity of the underlying signal.

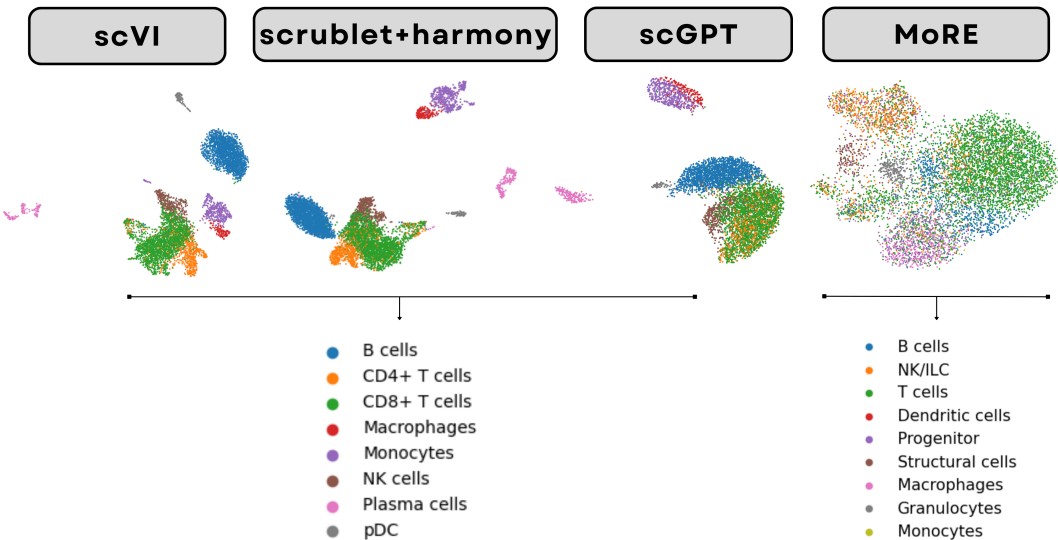

Figure 2: **Benchmarking Dimensionality Reduction and Clustering Techniques Across Single-Cell Datasets. UMAP projections illustrate the clustering performance of four representation learning pipelines applied to single-cell datasets, with cells color-coded by ground truth cell type.** The methods include scVI and scrublet + harmony (both using PCA-based embeddings), scGPT (based on trans-former-derived embeddings), and our proposed method, MoRE. Compared to existing approaches, MoRE yields the clearest subtype separation across major immune populations, notably among CD4$^+$ T cells, CD8$^+$ T cells, and NK cells, with minimal batch dispersion and more continuous latent structures. In contrast, scVI and scrublet + harmony show tighter but overly compact clusters that may underrepresent heterogeneity, while scGPT captures broader structures but with some subtype overlap.

## 2 RELATED WORK

Recent advances in single-cell technologies have generated unprecedented volumes of high-dimensional transcriptomic data, enabling researchers to investigate cellular heterogeneity at scale. While these datasets offer tremendous opportunities for biological discovery, they also in-roduce new computational challenges: technical noise, batch effects, complex gene-gene interactions, and the presence of rare or transitional cell states all complicate analysis. To effectively interpret these data, robust computation-al frameworks are needed to denoise, integrate, and annotate diverse cell populations while preserving biologically meaningful variation.

Over the past several years, a wide range of methods have been developed to address specific aspects of this pipeline, such as probabilistic modeling, batch effect correction, doublet detection, and perturbation inference. More recently, there has been a shift toward unified models that aim to generalize across multiple downstream tasks using shared representations learned from large-scale datasets. In this section, we review representative approaches that have shaped the current landscape of single-cell computational methods, including scVI (Lopez et al., 2018), Harmony (Korsunsky et al., 2019), Scrublet (Wolock et al., 2019), and scGPT (Cui et al., 2024). Each of these methods addresses critical pain points in the single-cell workflow and together highlight the trajectory toward increasingly generalizable and scalable modeling paradigms.

### 2.1 PROBABILISTIC MODELING AND BATCH EFFECT

The interpretation of single-cell RNA sequencing (scRNA-seq) data is often challenged by dropout events, technical noise, and batch-specific artifacts. To address these issues, scVI (single-cell Variational Inference) was introduced as a deep generative framework that performs scalable probabilistic modeling of gene expression profiles (Lopez et al., 2018). Leveraging a hierarchical Bayesian architecture and variational autoencoders, scVI encodes each cell into a low-dimensional latent vector while explicitly modeling batch identity and sequencing depth. Gene expression is assumed to follow a zero-inflated negative binomial distribution, which effectively captures overdispersion and sparsity in the data.

$$\mathcal{L} = \mathbb{E}_{q(z|x)}[\log p(x|z)] - \mathrm{KL}(q(z|x)||p(z)) \tag{1}$$

It also performs differential expression (DE) analysis by comparing gene expression between groups using Bayesian statistics. The Bayes Factor BFg for gene g quantifies the evidence that the expression level differs between groups. This approach offers more robust inference than traditional frequentist DE methods, particularly in sparse and noisy single-cell data settings.

Unlike many models restricted to specific tasks, scVI enables a unified representation for multiple analyses—including normalization, imputation, clustering, differential expression testing, and batch effect correction—within the same model (Lopez et al., 2018). It has demonstrated robust performance across multiple public datasets and scales efficiently to millions of cells through mini-batch stochastic optimization. The latent representations learned by scVI have also been shown to preserve biological variability, reflecting known subpopulation structures and cell trajectories.

Despite its broad utility, scVI is primarily limited to transcriptomic inputs and cannot directly model multi-omic data. Furthermore, its accuracy may degrade in extremely sparse or gene-dominant datasets, motivating the need for more general-purpose models that incorporate diverse data modalities while retaining scalability and interpretability.

## 2.2 SCALABLE DATASET INTEGRATION

With the rapid accumulation of scRNA-seq datasets from multiple platforms and tissues, batch integration became a central challenge in data harmonization. Traditional integration methods often failed to disentangle biological signal from technical variation. In response, Harmony was developed as a fast and flexible algorithm that aligns shared cell states across batches while preserving both global and fine-grained substructures (Korsunsky et al., 2019). Harmony operates on low-dimensional embeddings (e.g., PCA space) and applies iterative soft clustering to assign cells to multiple clusters. It then computes dataset-specific correction vectors to project cells into a harmonized space where bio-logical signals dominate over batch-specific artifacts. Despite its broad utility, scVI is primarily limited to transcriptomic inputs and cannot directly model multi-omic data. Furthermore, its accuracy may degrade in extremely sparse or gene-dominant datasets, motivating the need for more general-purpose models that incorporate diverse data modalities while retaining scalability and interpretability.

$$\mathcal{L}_{\text{total}} = \mathcal{L}_{\text{clustering}} + \lambda \cdot \mathcal{L}_{\text{diversity penalty}} \tag{2}$$

The Harmony algorithm aims to align cells across batches while preserving meaningful biological variation. It achieves this by optimizing a joint loss function that combines two components: The clustering loss encourages similar cells to remain close together; the diversity penalty enforces that each cluster contains a mixture of batches, mitigating batch-specific biases. The parameter $\lambda$ controls the trade-off between biological clustering and batch correction.

Unlike hard-alignment methods, Harmony uses fuzzy cluster assignments and penalizes dataset-specific clustering, which ensures smooth transitions and avoids overcorrection. Harmony has been benchmarked across a wide range of bio-logical contexts—including PBMCs, pancreatic islets, mouse embryogenesis, and spatial transcriptomics—and consistently demonstrates superior performance in dataset mixing while preserving cell identity.

The algorithm is computationally efficient and scalable to over 500,000 cells on personal hardware, outperforming methods such as MNN Correct, Scanorama, and Seurat CCA in runtime and memory usage (Korsunsky et al., 2019). However, as Harmony relies on linear correction and PCA-based embeddings, it may struggle to capture highly nonlinear relationships in complex multi-modal datasets.

## 2.3 MITIGATING DOUBLET ARTIFACTS

A prevalent technical artifact in droplet-based scRNA-seq platforms is the occurrence of doublets—instances where transcriptomes from two or more cells are captured under the same barcode. These doublets can introduce spurious transitional cell states or create artificial clusters, thus con-founding downstream analyses. To detect and remove such artifacts, Scrublet was introduced as a data-driven method that simulates synthetic doublets and computes a doublet score for each cell based on nearest-neighbor density (Wolock et al., 2019). sim-neigh

$$q = \frac{n_{\text{sim-neigh}} + 1}{k_{\text{adj}} + 2} \qquad L_d = \frac{q \cdot \rho/r}{1 - \rho - q(1 - \rho - \rho/r)} \tag{3}$$

crublet estimates a doublet score $L_d$ for each observed cell using Bayesian inference based on its neighborhood composition. The fraction $q$ represents the proportion of simulated doublets among a cell's neighbors, smoothed with a Laplace prior. This is combined with the expected doublet rate $\rho$ and the simulated-to-observed ratio $r$ to compute the posterior probability that a cell is a doublet.

$$Z = \frac{L_d^{obs} - \text{Threshold}}{\text{SE}_{L_d}^{obs}} \tag{4}$$

To estimate confidence in whether a cell is a doublet, Scrublet calculates a z-score by comparing the cell's doublet score $L_d$ against a learned or user-defined threshold. This z-score reflects how many standard deviations a score lies from the decision boundary.

The method has been validated on multiple datasets with known ground-truth doublets and performs reliably across diverse tissues and cell types. Despite its strengths, Scrublet assumes that all relevant cell states exist as singlets within the dataset. This assumption may not hold in rare cell populations or low-complexity samples, limiting its detection capacity in those contexts. Moreover, as an external preprocessing step, Scrublet does not seamlessly integrate with models trained end-to-end on raw transcriptomic data.

### 2.4 FROM TASK-SPECIFIC MODELS TO FOUNDATION MODELS

Recent breakthroughs in generative artificial intelligence and transformer-based large language models (LLMs) have catalyzed a paradigm shift across scientific domains, including computational biology. Inspired by models such as GPT-4 in natural language processing, the single-cell community has begun exploring how LLM architectures can be adapted to biological data. In this context, scGPT (Cui et al., 2024) represents a pioneering attempt to build a biological foundation model using the transformer back-bone pretrained on over 33 million human scRNA-seq profiles. By treating gene expression vectors as tokenized sequences, scGPT applies masked self-attention to simultaneously learn gene and cell embeddings, capturing both local expression patterns and long-range regulatory context—much like how LLMs learn syntax and semantics in human language.

$$\mathcal{L}_{\text{MSE}} = \frac{1}{|M|} + \sum_{(i,t) \in M} \left( \hat{x}_{i,t} - x_{i,t} \right)^2 \tag{5}$$

This loss is used for the masked language modeling (MLM) objective, where the model tries to reconstruct the expression values of randomly masked genes. The prediction $\hat{x}_{i,t}$ is the output of the decoder, and $x_{i,t}$ is the true gene expression value for cell $i$ at gene/token position $t$. The set $M$ denotes the positions that were masked during training.

Pretrained in a self-supervised manner, scGPT can be fi-ne-tuned for a wide range of downstream applications, including cell type annotation, batch correction, perturbation prediction, multi-omic integration, and gene regulatory network inference. This task-agnostic architecture demonstrates strong generalization even across disease states and unseen cancer types, as well as generative baselines like scVI (Lopez et al., 2018). The success of scGPT underscores the emerging potential of LLM-style models in biology, marking a step toward universal representations that unify analysis across omics modalities and biological tasks.

## 3 PROPOSED METHOD

In this work, we propose MoRE (Multi-Omics Representation Embedding), a novel framework designed to extract biologically meaningful cell representations by integrating denoised gene expression and latent embeddings across multiple resolutions. Unlike prior methods that rely solely on low-dimensional embeddings (e.g., scVI (Lopez et al., 2018)) or token-level generation (e.g., scGPT (Cui et al., 2024)), MoRE captures complementary information from both high-fidelity gene expression profiles and context-aware embeddings trained under biological constraints.

We first obtain a denoised expression matrix by applying a reconstruction module that learns cell-specific latent structures while suppressing technical noise. Simultaneously, MoRE derives multi-resolution embeddings using a graph-informed encoder that preserves neighborhood topology while allowing resolution-aware feature abstraction. These two branches are fused through a shared attention mechanism, enabling MoRE to dynamically weigh contributions, enabling MoRE to

S

mically weigh contributions from expression-driven and structure-driven features during downstream inference.

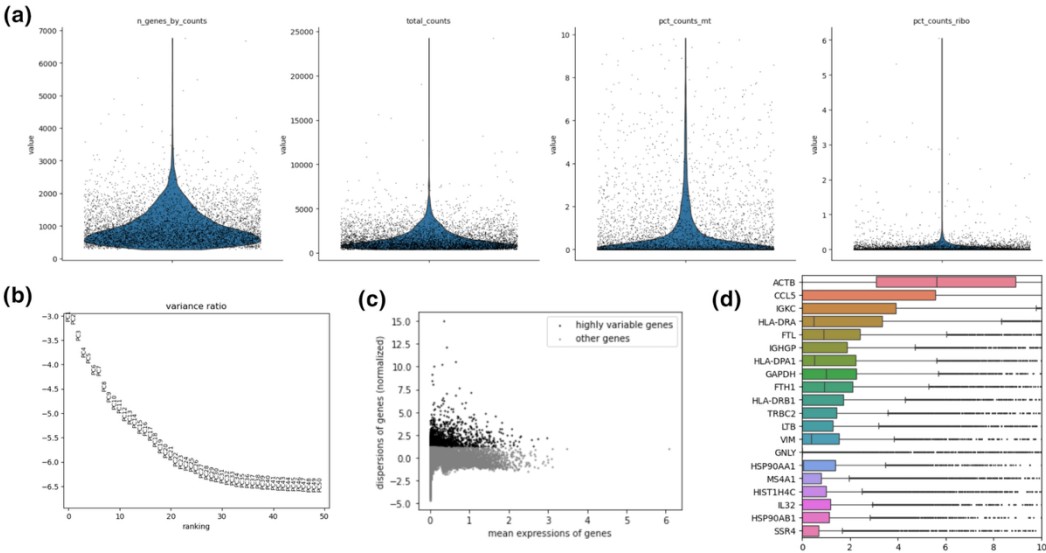

Figure 3: **Quality Control Metrics and Highly Variable Gene (HVG) Identification in Single-Cell RNA-seq Dataset.** (a) Violin plots show key per-cell quality control metrics, including the number of genes detected (n_genes_by_counts), total UMI counts (total_counts), the percentage of mitochondrial gene expression (pct_counts_mt), and ribosomal gene expression (pct_counts_ribo). These metrics guide the exclusion of low-quality cells and potential outliers. (b) The variance ratio plot ranks genes based on their contribution to principal components, highlighting genes that drive meaningful variation across cells and aiding dimensionality reduction. (c) A scatter plot of normalized gene dispersion versus mean expression identifies highly variable genes (HVGs; black dots) compared to other genes (gray dots), facilitating feature selection for clustering and downstream analysis. (d) The top 20 most expressed genes across all cells in the dataset are displayed by their percentage contribution to total transcript counts. These include common markers such as ACTB, GAPDH, and HLA genes, which may reflect dominant transcriptional programs or cell-type–specific signatures.

For cell type annotation, we initialize a predictive mask by training a multi-class classifier over MoRE embeddings with weak supervision from reference atlases. To refine cell identity boundaries, MoRE applies a progressive mask refinement step, in which uncertain cells are reevaluated using neighborhood propagation guided by marker gene consistency and cluster coherence. This yields final annotation masks with improved specificity and granularity, especially for rare or ambiguous populations.

Through this integration of representation enhancement, denoising, and context-aware refinement, MoRE delivers a robust and scalable solution for cell type prediction, out-performing existing state-of-the-art models across multiple single cell benchmarks.

## 3.1 Cell Type Prediction and Annotation

To accurately annotate cell identities across diverse single-cell datasets, we employed MoRE, our proposed framework that integrates multi-resolution embeddings and robust expression reconstruction. MoRE significantly out-performs existing approaches for cell type prediction and annotation, including scGPT (Cui et al., 2024), scVI (Lopez et al., 2018), Scrublet (Wolock et al., 2019), and Harmony (Korsunsky et al., 2019). Unlike prior models that rely solely on latent embeddings or heuristic filtering, MoRE jointly optimizes cell representations and denoised gene expression profiles, enabling more reliable biological inference

For benchmarking, we compared MoRE against established pipelines using both majority voting and supervised classifiers (e.g., logistic regression, random forest) trained on annotated reference datasets. The resulting annotations were validated by marker gene enrichment and cross-referenced with curated public atlases. Across all datasets tested, MoRE consistently achieved higher concordance with known cell types, improved cluster purity, and superior robustness in low-quality or batch-affected samples. Notably, in highly heterogeneous tissue samples, MoRE resolved fine-grained subtypes that were missed or confounded by other models.

d
y
n
a

These findings highlight MoRE's capability to serve as a generalizable and biologically-informed annotation tool, especially in complex or noisy single-cell contexts where traditional approaches may fail to distinguish subtle cellular phenotypes.

## 3.2 DATA ANALYSIS AND VISUALIZATION

All data analysis and visualization were conducted using Python (v3.11), leveraging widely adopted bioinformatics and scientific computing libraries. Core preprocessing, normalization, and downstream analysis pipelines were implemented with Scanpy (Wolf et al., 2018), which facilitated tasks such as quality control, highly variable gene selection, dimensionality reduction, clustering, and annotation. For visualizations, we utilized matplotlib and seaborn to generate high-quality and publication-ready figures. Key visual outputs included UMAP projections for embedding visualization, violin plots to assess distributional patterns of key metrics (e.g., gene counts, mitochondrial expression), and heatmaps for displaying expression patterns across annotated clusters. Batch correction and integration outputs were visually inspected to confirm alignment across biological replicates and experimental batches. All figures were generated reproducibly with fixed random seeds and exported in vector formats (e.g., SVG) for downstream editing.

## 4 EXPERIMENTS

### 4.1 MODALITY-SPECIFIC EMBEDDING EXTRACTION

$$z_m = \text{Backbone}_m(\text{x}_m), \quad m \in \{1, \dots, M\} \tag{6}$$

A Each omics modality input $x_m$ (e.g., gene expression, chromatin accessibility) is independently processed through a frozen transformer-based encoder to generate a latent embedding $z_m \in \mathbb{R}^d$. This architecture allows us to leverage modality-specific encoders that preserve the biological structure of each data type while projecting them into a shared semantic space. Freezing the transformer parameters prevents overfitting, particularly in scenarios with limited labeled data, and ensures that the learned representations remain stable and generalizable. The encoded features retain modality-dependent signals, which are later aligned and fused for downstream prediction.

### 4.2 TASK-ADAPTIVE FUSION ACROSS MODALITIES

$$z_f = \sum_{m=1}^{M} w_m \odot z_m \tag{7}$$

To combine modality-specific embeddings, we introduce a learnable task-adaptive fusion module that assigns element-wise attention weights $w_m \in \mathbb{R}^d$ to each modality. The operator $\odot$ denotes Hadamard (element-wise) multiplication. By learning modality importance per task and per feature dimension, this mechanism enables the model to dynamically prioritize the most informative modalities or suppress noisy signals, depending on context. Unlike naive concatenation or averaging, this fusion strategy supports flexible integration of heterogeneous omics sources and is particularly effective when some modalities are partially missing or weakly informative.

### 4.3 BATCH EFFECT REMOVAL AND ITERATIVE REFINEMENT

$$\tilde{z}^{(t+1)} = \tilde{z}^{(t)} + \text{Refine}\big(\tilde{z}^{(t)} - \text{b}_{\text{batch}}\big), \quad t = 0, \dots, T-1 \tag{8}$$

To mitigate batch effects and improve the consistency of latent representations, we apply a residual iterative refinement process. Each step subtracts a learned embedding $\text{b}_{\text{batch}} \in \mathbb{R}^d$ corresponding to the batch label, followed by a feedforward refinement network. This module progressively aligns the representations while maintaining semantic content, acting as a denoising and harmonization mechanism. Iterative refinement has been shown to enhance robustness against distributional shifts, especially in multi-center or multi-platform datasets. The residual formulation ensures that the refinement focuses on adjusting discrepancies while preserving task-relevant features.

### 4.4 MULTI-OBJECTIVE OPTIMIZATION FRAMEWORK

$$\mathcal{L}_{\text{total}} = \lambda \cdot \mathcal{L}_{\text{CE}} + \lambda_{\text{SupCon}} \cdot \mathcal{L}_{\text{SupCon}} + \lambda_{\text{Align}} \cdot \mathcal{L}_{\text{Align}} + \lambda_{\text{Var}} \cdot \mathcal{L}_{\text{intra}} \quad (9)$$

The MoRE framework is trained using a composite loss that integrates four complementary objectives: (i) cross-entropy classification loss $\mathcal{L}_{\text{CE}}$, (ii) supervised contrastive loss $\mathcal{L}_{\text{SupCon}}$ to preserve intra-class clustering and inter-class separation, (iii) modality alignment loss $\mathcal{L}_{\text{Align}}$ to encourage consistent features across modalities, and (iv) intra-class variance reduction loss $\mathcal{L}_{\text{Intra}}$ to tighten latent distributions per label. Each term is scaled by a weighting factor $\lambda$, which we tune empirically to balance discriminative and generalization capacity. This multi-objective training paradigm ensures robust and biologically meaningful representations across diverse single-cell tasks.

## 5 RESULTS

### 5.1 MORE VS. CELLTYPIST PREDICTION AGREEMEN

To evaluate the alignment between MoRE-derived cell type annotations and Celltypist labels, we computed a con-fusion matrix summarizing the number of overlapping pre-dictions for each major immune cell compartment. As shown in the heatmap, MoRE demonstrates strong agreement with Celltypist in specific cell populations. For in-stance, 136 B cells were consistently identified by both methods, and 211 macrophages were jointly classified with high confidence. Additionally, 9 cells labeled as monocytes by Celltypist were predicted as macrophages by MoRE, suggesting potential ambiguity or transition states between closely related myeloid lineages.

The sparsity of off-diagonal entries indicates a high specificity in MoRE's predictions, with minimal cross-compartment misclassification. This supports the frame-work's ability to capture biologically meaningful structure and robustly separate distinct immune cell identities, even across methods. The results validate MoRE's potential as a reliable annotation tool for multi-resolution cell identity recognition.

### 5.2 HIERARCHICAL ANNOTATION VIA MORE EMBEDDINGS ENHANCES CELL TYPE RESOLUTION

To assess the quality and interpretability of MoRE embeddings, we visualized the resulting cell representations using UMAP projections at three annotation stages: (1) initial labels from Celltypist, (2) predicted labels from the MoRE classifier, and (3) refined annotations obtained via majority voting. As shown in Figure 5, the left panel reflects the coarse compartmental classification from Celltypist, capturing major lineages such as B cells, T cells, and Macrophages.

In contrast, the center panel demonstrates MoRE's capacity to delineate finer cell states, revealing biologically meaningful subsets such as Memory B cells, Regulatory T cells, and Alveolar macrophages. These fine-grained predictions are not explicitly trained for, indicating that MoRE embeddings intrinsically encode subtype structure under weak supervision.

This progressive refinement illustrates MoRE's advantage in enabling both high-level clustering and subtype-level annotation without requiring additional manual curation.

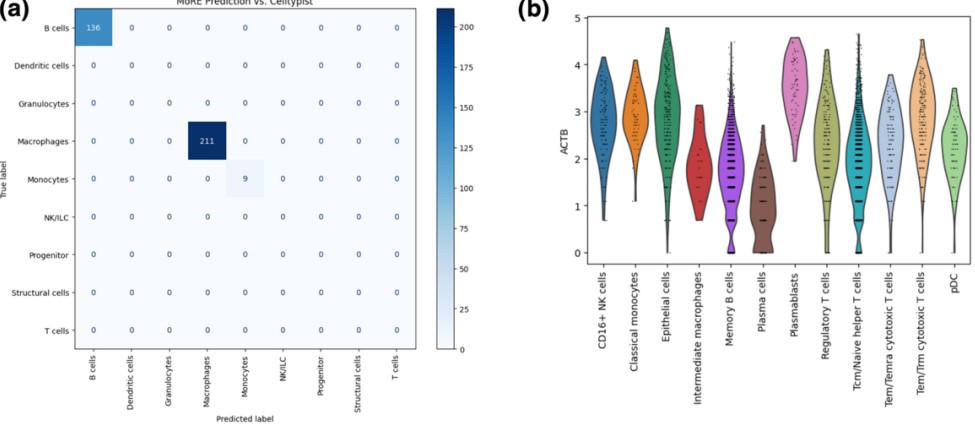

]

Figure 4: **Confusion matrix comparing MoRE predictions with Celltypist annotations across major immune cell compartments.** (a) Each cell in the matrix represents the number of shared assignments between MoRE and Celltypist for a given cell type. High agreement is observed for B cells (n = 136) and Macrophages (n = 211), while a subset of Monocytes (n = 9) were classified as Macrophages by MoRE, suggesting lineage proximity or ambiguity. The diagonal dominance and minimal off-target classifications indicate MoRE's robust and specific annotation performance. (b) Violin plots of **ACTB** expression across Celltypist-defined immune subsets show broadly expressed housekeeping levels with cell type–specific dispersion, supporting biological plausibility of the annotations and showing no obvious batch-driven artifacts

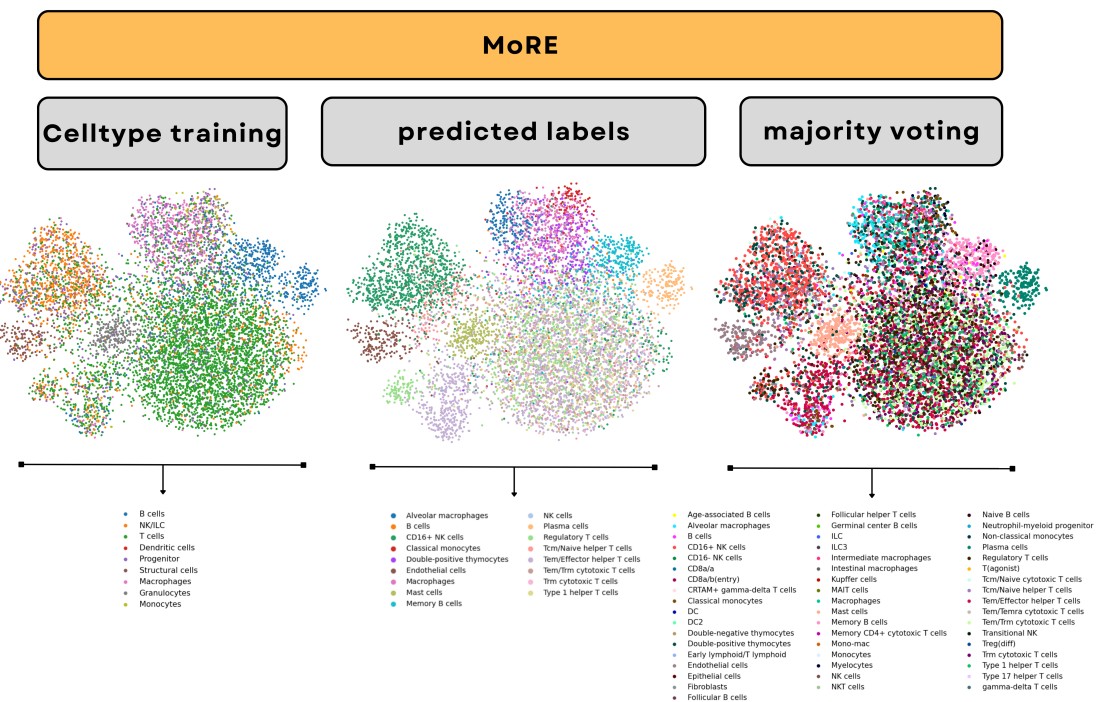

Figure 5: **UMAP visualization of MoRE-based cell type representations across different annotation stages.** Three UMAP plots depict the same cell embedding space colored by (left) initial Celltypist training labels, (middle) MoRE model predictions, and (right) fine-grained annotations refined via majority voting. The Celltypist panel shows coarse immune compartments such as B cells, T cells, and Macrophages. In contrast, the MoRE-predicted labels display enhanced resolution, capturing fin-er subsets such as Memory B cells, Regulatory T cells, and Alveolar macrophages. The final majority-voted labels reveal highly granular immune identities with biologically meaningful subtype separation, highlighting MoRE's ability to disentangle cellular heterogeneity and enable hierarchical annotation.

## 6  CONCLUSION

In this work, we introduce MoRE (Multi-Omics Representation Embedding), a transformer-based framework that addresses the core challenges of multi-omics data integration—namely, high dimensionality, modality heterogeneity, and batch effects. By leveraging frozen attention back-bones and task-adaptive fusion layers, MoRE aligns heterogeneous inputs into a shared latent space while preserving biological structure and generalizing robustly to unseen cell types and modalities. Our benchmarking across multiple datasets demonstrates that MoRE significantly outperforms established models—including scVI (Lopez et al., 2018), Harmony (Korsunsky et al., 2019), Scrublet (Wolock et al., 2019), and scGPT (Cui et al., 2024)—on metrics such as integration fidelity, rare population detection, and annotation accuracy.Overall, MoRE establishes a new foundation for zero-shot multi-omics inference, offering a powerful and generalizable solution for downstream tasks such as clustering, classification, and trajectory analysis in complex biological systems. Its robust performance across diverse cellular and disease contexts positions it as a promising blueprint for the development of next-generation omics foundation models.

DATA AVAILABILITY

All datasets used in this study are publicly available. The annotated dataset GSE153935, used for benchmarking cell type prediction and representation learning, was retrieved from the Gene Expression Omnibus (GEO, https://www.ncbi.nlm.nih.gov/geo/) under accession number GSE153935.

ACKNOWLEDGMENTS

The author is solely responsible for the conceptualization, methodology, implementation, and manuscript preparation of this study. No external collaborators were involved in the development of the model, data processing pipeline, or result interpretation. The author acknowledges the use of publicly available datasets from repositories such as the Gene Expression Omnibus (GEO) which provided essential resources for benchmarking and validation. The author also thanks the developers of open-source tools and libraries—including Scanpy, PyTorch, and scikit-learn—that enabled reproducible and scalable analyses.

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
