# OpenReview forum: "MoRE: Batch-Robust Multi-Omics Representations from Frozen Language Models"
_ICLR.cc/2026/Conference — ICLR 2026 Conference Withdrawn Submission_

### Official Review · Reviewer_TH3o · 2025-10-28

**Soundness:** 1
**Presentation:** 1
**Contribution:** 1
**Rating:** 2
**Confidence:** 5

**Summary:**

This paper introduces MoRE, a framework for learning batch-robust multi-omics representations. The method repurposes a frozen, pre-trained transformer backbone—originally developed for biological sequence data—as a feature extractor for single-cell omics profiles. Lightweight, task-specific adapters are then trained to project different omics modalities into a shared embedding space, which is further refined using a combination of supervised contrastive learning and alignment losses to mitigate batch effects. The authors claim that this approach enables effective integration across modalities and improves generalization to unseen cell types and platforms. Evaluations against methods like scGPT and scVI are presented to support its performance in tasks such as cell type annotation and batch correction.

**Strengths:**

1. Motivating a Challenging Problem: The work tackles the relevant and non-trivial challenge of integrating heterogeneous multi-omics data while correcting for batch effects, a significant pain point in single-cell genomics.

2. Conceptual Proposition: The idea of leveraging a frozen, pre-trained model backbone to create a stable, general-purpose feature extractor is a conceptually interesting direction for improving model generalizability and computational efficiency.

3. Modular Design Intent: The proposed framework suggests a modular architecture, which, in principle, could offer flexibility for incorporating new data modalities and tasks.

**Weaknesses:**

1. Misleading Terminology and Presentation: The persistent use of the term "Frozen Language Models" is highly misleading, strongly implying the use of natural language text and models like GPT. In reality, the method merely employs a transformer architecture on biological sequences, constituting a significant misrepresentation of the work's actual technical basis.

2. Insufficient and Unconvincing Empirical Validation: The experimental evaluation fails to substantiate the paper's central claims. There is a notable absence of quantitative benchmarks on the key promised tasks, such as batch effect correction (e.g., using metrics like ASW or LISI) or multi-omics integration. The comparison with Celltypist is not a rigorous validation of representation quality but rather an agreement study between two classifiers.

3. Unsubstantiated Claims of Superiority: The paper repeatedly claims to outperform strong baselines (e.g., scGPT, scVI, Harmony) but provides minimal quantitative evidence to support these assertions. The results presented do not demonstrate a clear or significant advantage over existing methods.

4. Poorly Defined Novelty and Contribution: The core technical approach of using a frozen pre-trained backbone with lightweight adapters is a well-established paradigm in transfer learning, not a novel innovation. The paper fails to articulate a specific, meaningful advancement beyond the current state-of-the-art, making its contribution ambiguous.

5. Disconnect Between Claims and Evidence: A significant gap exists between the ambitious claims made in the abstract and introduction (e.g., "zero-shot generalization," "batch-robust," "practical scalability") and the limited, tangential evidence provided in the results section. The work remains largely aspirational.

6. Severe Structural and Organizational Deficiencies: The paper exhibits a critical failure in logical organization. Section 3 ("Proposed Method") is cluttered with implementation details that belong in an "Experimental Setup," while Section 4 ("Experiments") confusingly introduces core methodological components (e.g., the embedding extraction and fusion modules). This profound disjunction between the high-level framework narrative and the low-level technical description makes it impossible to cleanly separate the proposed model's conceptual innovation from its specific instantiation.

**Questions:**

1. Your abstract claims superiority in "integration fidelity, rare population detection, and modality transfer." Could you provide quantitative results on standard benchmarks for these tasks, such as batch integration scores (e.g., ASW, LISI) or metrics for rare cell detection, to substantiate these claims?

2. The term "Frozen Language Models" strongly implies the use of models trained on natural language. Given that you are processing biological sequences, do you agree that this terminology is potentially misleading and should be revised to more accurately reflect the technical approach (e.g., "frozen transformer backbone")? If not, please provide more evidence to support your opinion.

3. The manuscript lacks sufficient implementation details to ensure reproducibility. Could you please provide a comprehensive description of the training configuration, including key hyperparameters (e.g., learning rate, batch size, optimizer), the specific dimensions of the fusion modules and adapters, and the precise stopping criteria for the iterative refinement process?

4. What is the specific purpose of Figure 3 in the context of validating the MoRE framework, as it is not referenced in the main text? Furthermore, the description of data preprocessing and quality control is cursory. Please elaborate on the exact filtering thresholds applied (e.g., for gene counts, mitochondrial percentage) and the methodology for Highly Variable Gene (HVG) selection.

5. Figure 4a seems counterintuitive. Taking T cells as an example, the rows of True labels are all zero, and the columns of Predicted labels are also all zero. So why do T cells appear in this heatmap? Furthermore, this heatmap shows that the dataset only contains 136 + 211 + 9 = 356 cells, which is too few. Finally, a more precise description of this figure is needed. What are the True labels? Are they the cell types described by experts, or are the cell types annotated by Celltypist considered the truth?

6. The process for cell type annotation is completely inconsistent with consensus. Celltypist is simply another annotation method and should not be considered ground truth. The appropriate process is to select a dataset with expert-annotated cell types and consider this expert annotation as ground truth. Then, use methods such as MoRE, scGPT, and Celltypist to annotate cell types. Finally, use the F1 score or accuracy to evaluate the consistency of the predicted results with the expert annotations.

7. The biological insight intended from Figure 4b (ACTB expression) is unclear, as it is not discussed in the main text. What specific conclusion regarding the model's performance or the data's biology should the reader draw from the expression profile of this ubiquitous housekeeping gene?

8. Section 5.2 claims that MoRE can discern finer-grained cell states than the initial Celltypist labels. What independent, quantitative evidence—such as marker gene enrichment or concordance with known cellular hierarchies—can be provided to validate the biological accuracy of these postulated subtypes, beyond visual separation in UMAP space?

---

### Official Review · Reviewer_B24R · 2025-10-30

**Soundness:** 1
**Presentation:** 1
**Contribution:** 1
**Rating:** 2
**Confidence:** 4

**Summary:**

This paper proposes MoRE, a framework that repurposes a frozen transformer backbone plus lightweight, modality-specific adapters and a task-adaptive fusion layer to produce batch-robust multi-omics embeddings. The paper claims generalization across unseen batches, donors, platforms, and modalities on single-cell tasks.

**Strengths:**

1. Recycling a frozen transformer with modality-specific adapters is a clear high-level idea.

2. The freeze-the-backbone approach could reduce compute and mitigate catastrophic forgetting during onboarding of new modalities.

**Weaknesses:**

1. Despite strong claims, the paper does not report standard, widely accepted measures for batch effect removal and biological conservation, such as kBET, ASW, graph iLISI, kNN-mixing scores, or NMI for label structure. Without these, batch-robust is not substantiated.

2. Figure 2 is not evidence of batch correction or superior embeddings. A single UMAP is insufficient to claim batch removal and one must at least show color-by-batch overlays and quantify mixing. Moreover, Figure 2 appears to lose a distinct B-cell cluster, undermining the structure-preserving claim. Clustering quality should be reported with NMI, F1, ASW, and stability.

3. Although the method is presented as multi-modal, all downstream experiments are on scRNA-seq only. To support the central thesis, the paper must include bona fide multi-omics datasets and tasks (e.g., 10x Multiome RNA+ATAC, SHARE-seq, SNARE-seq, CITE-seq RNA+ADT), including modality transfer and missing-modality scenarios.

4. Comparisons exclude canonical multi-omics integration methods such as GLUE, MultiVI, Seurat, MOFA and more recent cross-modal models. Since the paper’s innovation overlaps with multi-omics representations, not comparing to these makes the empirical case unconvincing.


5. Ablations and diagnostics are absent (e.g., adapters vs. no adapters and each loss component).

**Questions:**

Please see Weaknesses.

---

### Official Review · Reviewer_8iKx · 2025-11-01

**Soundness:** 1
**Presentation:** 1
**Contribution:** 2
**Rating:** 0
**Confidence:** 4

**Summary:**

This work introduces MoRE (Multi-Omics Representation Embedding), a framework to extract biologically meaningful cell representations by integrating modality-specific embeddings for different omics which are projected into a shared semantic space. MoRE is benchmarked against established single-cell models and methods, including scVI, Harmony, Scrublet, scGPT, and CellTypist across multiple tasks and datasets, with promising initial results.

**Strengths:**

•	This paper introduces a novel method in an underexplored and important area
•	Promising results across various tasks.

**Weaknesses:**

•	A critical flaw of this paper is that it makes the claim:


 "Our benchmarking across multiple datasets demonstrates that MoRE significantly outperforms established models—including scVI (Lopez et al., 2018), Harmony (Korsunsky et al., 2019),Scrublet (Wolock et al., 2019), and scGPT (Cui et al., 2024)—on metrics such as integration fidelity, rare population detection, and annotation accuracy."

However, those results do not appear to be in the manuscript. While the models listed above are evaluated on clustering performance and compared against MoRE, and MoRE is also evaluated on other tasks such as annotation accuracy and compared against CellTypist, these results are not sufficient in supporting the paper's main claims, and providing a comprehensive view of MoRE’s performance.
•	A minor flaw is that tables are often not referred to in the text, detracting from clarity (ie. see section 5.1, section 3.1).
•	The paper is poorly organized - the introduction is a mix of different sections that include results, discussion, background and significance, and the text traverses back and forth among these sections, creating a confusing narrative and leaving the reader unclear about any of the paper claims.
•	The rationale for selection of the baselines (claimed as strong baselines) is lacking. Why include scrublet, which is a QC method?
•	The MoRE method is insufficiently described (just a few sentences are provided). At least an architecture diagram would be needed.

This paper omits key results in supporting its main claims. I am therefore providing an initial recommendation of rejection.

**Questions:**

•	Can the authors compare MoRE’s performance on integration fidelity, rare population detection, and annotation accuracy against all the models it was claimed MoRE was compared to?
Some suggestions:
•	Consider moving minor details such as figure generation to an appendix.
•	Ensure that tables are referred to in the text where appropriate.
•	There are a significant number of typos in the paper. Please further edit the manuscript (for example, the title for section 5.1 has a typo).

---

### Official Review · Reviewer_v5nc · 2025-11-01

**Soundness:** 2
**Presentation:** 2
**Contribution:** 2
**Rating:** 2
**Confidence:** 5

**Summary:**

This paper presents MoRE, an LLM-inspired framework for multi-omics representation embedding. MoRE repurposes frozen transformer backbones augmented with lightweight, modality-specific adapters and a task-adaptive fusion layer to map heterogeneous single-cell omics into a shared latent space. Experimental results show that MoRE significantly outperforms scGPT, scVI, Scrublet, and Harmony in integration fidelity, rare-population detection, and cross-modality transfer.

**Strengths:**

1. The combination of a frozen backbone, parameter-efficient fusion, and iterative batch-effect refinement provides a clear cross-modality alignment scheme.
2. Compared with prior generative-reconstruction approaches, it places greater emphasis on structure-preserving alignment, which is relatively novel.

**Weaknesses:**

1. This manuscript uses the wrong template. For example, it is clearly missing line numbers.
2. There is a GitHub repo link in the abstract, which might be a violation of the double-blind policy.
3. The paper lacks ablation studies to demonstrate the necessity of each module.
4. The paper has typos and a confusing overall presentation. For example, the Proposed Method and Experiments sections seem to be in the wrong order.
5. The figures need better quality. For instance, the description of Figure 1 is unclear. Figure 4 also has a layout problem: its caption should not be separated from the figure or split across pages.
6.The training setting needs further clarification, including which components are frozen and which are trained. The reproducibility details, such as data splits and random seeds, are incomplete.

**Questions:**

1. In the Introduction, the paper describes the approach as “training-free” for multi-omics integration. However, the modality-specific adapters and the task-adaptive fusion layer need to be trained. The authors should reconcile this and improve the wording.
2. Please check Equation (5). The "+" shall be redundant.
3. Please provide ablation results that clarify the incremental contributions of iterative refinement and dense alignment beyond simple fusion.
4. Please provide a detailed description of the domain splits and missing-modality configurations, along with the corresponding quantitative results.

---

### Note · Authors · 2025-11-12

I have read and agree with the venue's withdrawal policy on behalf of myself and my co-authors.